# The Chemopreventive Effects of Chlorogenic Acids, Phenolic Compounds in Coffee, against Inflammation, Cancer, and Neurological Diseases

**DOI:** 10.3390/molecules28052381

**Published:** 2023-03-04

**Authors:** Toshiyuki Murai, Satoru Matsuda

**Affiliations:** 1Graduate School of Medicine, Osaka University, 2-2 Yamada-oka, Suita 565-0871, Japan; 2Department of Food Science and Nutrition, Nara Women’s University, Kita-Uoya Nishimachi, Nara 630-8506, Japan

**Keywords:** polyphenol, chlorogenic acid, inflammation, cancer progression, neurodegenerative diseases, cell membrane

## Abstract

Coffee is one of the most widely consumed beverages, which has several effects on the human body. In particular, current evidence suggests that coffee consumption is associated with a reduced risk of inflammation, various types of cancers, and certain neurodegenerative diseases. Among the various constituents of coffee, phenolic phytochemicals, more specifically chlorogenic acids, are the most abundant, and there have been many attempts to utilize coffee chlorogenic acid for cancer prevention and therapy. Due to its beneficial biological effect on the human body, coffee is regarded as a functional food. In this review article, we summarize the recent advances and knowledge on the association of phytochemicals contained in coffee as nutraceuticals, with a particular focus on phenolic compounds, their intake, and nutritional biomarkers, with the reduction of disease risk, including inflammation, cancer, and neurological diseases.

## 1. Introduction

Coffee is one of the most widely consumed beverages globally [1]. Early records suggest that coffee was first discovered and consumed in Ethiopia, North Africa, in the 9th century AD [2]. Coffee plants belong to the genus *Coffea*, among which *Coffea arabica* and *Coffea canephora* var. Robusta, also known as Arabica and Robusta coffee, respectively, are the most well-known species [3]. Coffee beans are roasted, ground, and infused in hot water to produce a cup of coffee. In addition to its pleasantly bitter flavor, coffee has several effects on the human body and mind [4].

Coffee is popularly consumed as a caffeine-containing beverage (CCO) and has potential health benefits and risks based on the food-based dietary guidelines of the Food and Agriculture Organization (FAO) of the United Nations [5]. Epidemiological studies have demonstrated that coffee consumption reduces the risk of neurological diseases, including Alzheimer’s disease and Parkinson’s disease [6,7,8]. For example, coffee consumption ameliorated cognitive impairment induced by Alzheimer’s disease [7]. Another report showed a negative association between moderate consumption of coffee and the risk of age-related cognitive disorders and Parkinson’s disease [8]. Coffee consumption has been also linked to potential health benefits as a consequence of their chemopreventive and anti-inflammatory effects. It has been suggested that the reduction of inflammation under administration of coffee is attributed to the antioxidative features of certain ingredients of coffee [9]. Furthermore, current evidence suggests that coffee consumption is associated with a reduced risk of liver, kidney, and to a lesser extent, premenopausal breast and colorectal cancers, while it is unrelated to prostate, pancreas, and ovary cancers [10]. Although there are several plausible biologic mechanisms whereby coffee consumption might influence the risk of breast cancer, epidemiologic evidence is limited [11]. Meanwhile, Nkondjock et al. assessed the association between coffee consumption and breast cancer risk among high-risk women carrying breast cancer susceptibility genes (BRCA) mutations, and the results suggested that coffee is not only unlikely to be harmful but also high levels of coffee consumption may be linked to reduced breast cancer risk [12]. Many studies have demonstrated the relationship between coffee consumption and cancer risk [13,14,15,16,17,18,19,20,21,22,23,24,25,26]. Therefore, in the present review, we summarize the recent advances and knowledge on the associations of phytochemicals present in coffee as nutraceuticals, particularly focusing on phenolic compounds in coffee, their intake, and nutritional biomarkers with reduction of disease risk including inflammation, cancer, and neurological diseases.

## 2. Chemical Ingredients of Coffee

Coffee is a major dietary source of purine alkaloid caffeine (1,3,7-trimethylxanthine; 1,3,7-trimethyl-1*H*-purine-2,6-(3*H*,7*H*)-dione), which exerts various effects via the A1 and A2 adenosine receptor subtypes, effectively stimulating the sympathetic nervous system. However, caffeine intake also has negative effects although the amount of caffeine in a cup of coffee is influenced by the method of coffee preparation such as boiled coffee, filtered coffee, and espresso, and the half-life of caffeine in the human body is approximately 4–6 h [9]. Besides caffeine, coffee is a rich source of various phytochemicals naturally present as secondary plant metabolites. Coffee is abundant in several phenolic compounds, among which are chlorogenic acid and caffeic acid, lactones, diterpenes including cafestol and kahweol, and the niacin (vitamin B_3_) precursor trigonelline [27,28]. Particularly, green coffee contains various kinds of phenolic compounds accounting for 6–10% of its dry weight [5]. Chlorogenic acid, a type of acyl-quinic acid, i.e., a family of 1L-(−)-quinic acid esters combined with C6−C3 trans-hydroxycinnamic acid, is the major phenolic compound in coffee [9], which comprises three groups of chemical compounds, namely caffeoylquinic acids, feruloylquinic acids, and dicaffeoylquinic acids (Figure 1). The main components of the coffee polyphenols are caffeoylquinic acids. Coffee caffeoylquinic acids comprise three chemical compound isomers, and the most common form of chlorogenic acid is 5-*O*-caffeoylquinic acid, which is often called chlorogenic acid [29,30]; chlorogenic acid (5-*O*-caffeoylquinic acid; 5-CQA), neo-chlorogenic acid (3-*O*-caffeoylquinic acid; 3-CQA), and crypto-chlorogenic acid (4-*O*-caffeoylquinic acid; 4-CQA) (Figure 1). The chlorogenic acids found in green coffee beans contain 3-, 4-, and 5-caffeoylquinic acids and 3,4-, 3,5- and 4,5-caffeoylquinic acids, collectively referred to as total caffeoylquinic acids, and the composition depends on the type of coffee. They also contain 3-, 4-, and 5-feruloyl quinic acids and traces of at least one caffeoyl-feruloyl quinic acid, also known as total feruloyl quinic acids [31,32]. The chemical structures are shown in Figure 1. Among these chlorogenic acids, 5-CQA is the most abundant in coffee beans, accounting for approximately 50% of the total chlorogenic acids [33,34,35]. There are many attempts to utilize coffee chlorogenic acid for cancer prevention and therapy [36].

## 3. The Metabolism of Chlorogenic Acids

The biosynthesis of chlorogenic acids in humans is mainly mediated by three key enzymes, namely phenylalanine ammonia-lyase, shikimic acid/quinic acid hydroxyl cinnamyl transferase, and quinic acid cinnamate hydroxyltransferase [37]. The first enzyme, phenylalanine ammonia-lyase, acts as a rate-limiting enzyme of the chlorogenic acid biosynthetic pathway and catalyzes the dissociation reaction of an ammonia molecule from an l-phenylalanine to produce a *trans*-cinnamic acid. The second enzyme, quinic acid hydroxyl cinnamyl transferase, catalyzes the formation of p-coumaryl-quinic acid/shikimic acid, while quinic acid cinnamate hydroxyltransferase catalyzes the transesterification of cafeyl-CoA and quinic acid to generate chlorogenic acid. Dietary chlorogenic acids are hydrolyzed into quinic and caffeic or ferulic acid, and then further metabolized in the small intestine and colon before entering the bloodstream [37] (Figure 2). Caffeic acid, e.g., 3, 4-dihydroxycinnamic acid, is converted by the enzyme catechol-*O*-methyltransferase to another phenolic acid, ferulic acid. Both compounds may form an ester bond with quinic acid, and generate any of the many isomers included in the family of chlorogenic acids. Nonetheless, the most frequent isomer is the 5-*O*-caffeoylquinic acid that, because of that, is commonly called chlorogenic acid.

## 4. Anti-Inflammatory Activity of Chlorogenic Acids

Reactive oxygen species (ROS) cause oxidative stress that contributes to the pathogenesis of various diseases, including inflammation, cancer, and neurodegenerative diseases [38,39]. ROS, including radical and non-radical derivatives, such as superoxide anions, hydroxyl radicals, and hydrogen peroxide, mainly derived from oxidative metabolism during inflammatory reactions [38], are generated by redox reactions during cellular metabolism. However, excess production of ROS can cause oxidative damage to essential molecules such as proteins, lipids, and DNAs [39]. Chlorogenic acids exert their antioxidant effect via their polyhydroxyl structure that directly scavenges free radicals and regulates the activity of the endogenous oxidase system and its associated proteins [40,41]. This natural antioxidant property depends on the chlorogenic acid’s unique molecular structure, which contains several active hydroxyl groups and one carboxyl group. Of these, the phenolic hydroxyl structure readily reacts with free radicals and forms hydrogen free radicals, which eliminate hydroxyl radicals and superoxide anions and exhibit a strong antioxidant effect [37,42]. Thus, under some circumstances, coffee might contribute to the endogenous systems which prevent oxidative damage to cell components, DNA, proteins, and lipids, which contribute to the pathogenesis of inflammation, cancer and neurodegenerative diseases [43]. Consequently, the consumption of instant coffee, which contains increased levels of chlorogenic acids, enables protection against oxidative damage in healthy adults [44].

Chlorogenic acids can eliminate superoxide anions and hydroxyl radicals through their antioxidant activities, rendering coffee an effective dietary antioxidant source due to its high chlorogenic acid content [45]. Chlorogenic acids directly act on the nuclear factor kappa-light-chain-enhancer of activated B cells (NF-κB) signaling pathway and control the expression of both pro- and anti-inflammatory factors [46,47]. Studies have shown that chlorogenic acids can inhibit interleukin-8 (IL-8) production in human intestinal Caco-2 cells, induced by combined stimulation with tumor necrosis factor-alpha (TNF-α) and H_2_O_2_ [48]. IL-8 is a cytokine similar to platelet factor 4, with a chemoattractive activity. IL-8 is produced by phagocytes and mesenchymal cells exposed to inflammatory stimuli and activates neutrophils inducing chemotaxis. These results suggest that dietary chlorogenic acids might prevent intestinal inflammation [48]. Vascular endothelial cells exhibit upregulation of adhesion molecules, such as intercellular adhesion molecule 1 (ICAM-1) and vascular cell adhesion molecule 1 (VCAM-1), which enable recruitment of immune cells, including monocytes, to the inflammation site. ICAM-1 and VCAM-1 are the members of the immunoglobulin superfamily of transmembrane adhesion molecules with an amino-terminus extracellular domain, a single transmembrane domain, and a carboxy-terminus cytoplasmic domain. VCAM-1 binds to its ligand, very late antigen-4 (VLA-4). Furthermore, chlorogenic acid attenuates the enhanced expression of ICAM-1 and VCAM-1 induced by interleukin-1β, highlighting its anti-inflammatory activities [49]. The recruitment of neutrophils to the site of inflammation is a typical process during inflammation, and the interactions of the circulating neutrophils with the vascular endothelial cells is the initial step of neutrophil recruitment, which is mediated by certain adhesion molecules expressed on the surface of neutrophils. In this context, an adhesion molecule CD62L, also called L-selectin, plays a pivotal role in cell–cell interactions. CD62L is a member of the carbohydrate-binding selectin family of cell adhesion molecules and forms a type-I transmembrane protein comprising an N-terminal lectin domain, an epidermal growth factor (EGF)- like domain, two short consensus repeats, a transmembrane region, and a short cytoplasmic domain. CD62L is expressed in leukocytes and located at the cell surface, which plays a pivotal role in multistep cell-cell adhesion interactions. A previous study reported that chlorogenic acid could attenuate the lipopolysaccharide-induced CD62L proteolytic processing of neutrophils and decrease the adhesion and chemotaxis of neutrophils to vascular endothelial cells [50]. Platelets are also key mediators of inflammation and platelet–endothelial cell interactions at the site of lesion trigger inflammatory responses. Upon platelet activation, the CD62P adhesion molecule, also called P-selectin, translocates from the Weibel–Palade bodies inside the cell body into the plasma membrane, where fibrinogen causes platelet aggregation. In addition, chlorogenic acid inhibits the expression of CD62P in human platelets and impair platelet–leukocyte interactions [51]. 

## 5. Anti-Cancer Activity of Chlorogenic Acids

Numerous epidemiological studies have indicated that coffee consumption might lower the risk of certain types of cancer. For instance, coffee consumption has been found to strongly and consistently reduce the risk of endometrial and hepatocellular cancer [10], and a modest or borderline negative association with breast and colorectal cancer has been reported. Contrastingly, no association was found with pancreatic, ovarian, prostate, or gastric cancer [10]. The epidemiologic evidence of coffee on each type of cancer is summarized in the literature [52].

One of the eight distinct hallmarks of cancer involves the acquired capability for sustaining proliferative signaling [53,54]. The first step of cancer progression toward poorly differentiated carcinomas is dedifferentiation which is not initially associated with increased proliferation. A study showed that chlorogenic acids could inhibit the proliferation of A549 human lung cancer cells in vitro by inhibiting activator protein-1, NF-κB, and mitogen-activated protein kinases (MAP kinases) [55]. NF-κB plays critical roles in inflammation, cell proliferation, differentiation, and survival. MAP kinases have three main families, extracellular-signal-regulated kinases (ERKs), jun amino-terminal kinases (JNKs), and p38/stress-activated protein kinases (SAPKs). These respond primarily to growth factors and mitogens to induce cell growth and differentiation. This suggests that the consumption of chlorogenic acids through coffee might prevent cancer [56,57]. 

In addition, matrix metalloproteinases (MMPs) are essential enzymes employed by tumor cells during metastasis that degrade proteins and regulate various cell behaviors [58,59]. These proteolytic enzymes are prevalent in cancer biology due to their capacity to promote cancer-cell growth, differentiation, apoptosis, migration, and invasion while they also regulate tumor angiogenesis and immune surveillance [58]. The MMPs belong to a family of zinc-dependent endopeptidases with more than 20 different members, and play pivotal roles in the degradation of the extracellular matrix which is composed of collagens, fibronectins, and laminins, which help maintain homeostasis. Based on their sub-cellular distribution and specificity for components of the extracellular matrix, the MMPs are divided into collagenases, gelatinases, stromelysins, matrilysins, and membrane-type matrix metalloproteases: MMPs that belong to collagenases are MMP-1, MMP-8, MMP-13, and MMP-18, which degrade triple-helical fibrillar collagen; MMPs that belong to gelatinases are MMP-2 and MMP-9; stromelysins include MMP-3 and MMP-10; MMP-11, and MMP-7 and MMP-26 are matrilysins. MMPs are inhibited by endogenous protein regulators, namely, the tissue inhibitors of MMPs (TIMPs). Most MMPs have consistently increased gene expression across cancers, and MMP1, MMP9, MMP10, MMP11, and MMP13 are almost universally upregulated across a wide variety types of cancers [59]. Of all MMPs, MMP-9 is the most essential for cancer-cell invasion and tumor metastasis [60]. Chlorogenic acids have been shown to inhibit MMP-9 activity in cultured hepatoma cells, indicating a possible cancer chemoprevention mechanism [61]. Furthermore, MMP-2, also called gelatinase A, also plays a significant role in ECM degradation because it can degrade collagen type IV during cancer progression, allowing cancer cells to migrate from the primary tumor to form metastasis [62]. However, chlorogenic acids have been found to inhibit cell migration and MMP-2 secretion of human glioma cells, highlighting their anti-cancer effects [63].

The signaling pathway comprising phosphatidylinositol 3-kinase (PI3K)/AKT/mammalian target of rapamycin (mTOR)/phosphatase and tensin homologue deleted on chromosome ten (PTEN) plays a pivotal role in cancer progression, including cell proliferation and migratory activities [64]. Particularly, PI3K has been shown to induce the expression of the multidrug resistance-associated protein, suggesting that high PI3K activity facilitates drug resistance [64]. Therefore, the PI3K/AKT/mTOR/PTEN axis is an attractive target for targeted molecular therapy including cancer [65,66,67,68,69,70,71]. Moreover, germline mutations in the breast cancer susceptibility gene 1 (BRCA1) considerably increase the risk of breast and ovarian cancers and, thus, modulation of PTEN/BRCA1 proteins may prove therapeutically beneficial for breast, ovarian, and prostate cancer treatment [72]. Chlorogenic acids have a biological activity to modulate signal transduction through the PI3K/AKT/PTEN pathway, thereby suppressing cancer progression. Chlorogenic acid potentiated the apoptotic effect of certain anti-cancer agents via activation of apoptosis-related molecules, namely Bcl-2-associated X protein (Bax) and Caspase 3/7, and inhibition of anti-apoptotic molecules, namely B-cell/CLL lymphoma-2 (Bcl-2) and B-cell lymphoma-extra large (Bcl-xL), by modulating the PI3K/Akt signaling pathway [73]. Moreover, chlorogenic acid could selectively suppress the proliferation of human kidney cells by modulating the PI3K/Akt/mTORC signaling pathway [74]. 

Wnt signaling is another signaling pathway that plays an important role in cancer-cell signaling. The Wnt signal is transferred into the cytosol and thereby further transduced to the cell nucleus via the β-catenin–T-cell factor and lymphoid enhancer factor (TCF/LEF) complex to enhance the expression of targets, including Myc and leucine-rich repeat-containing G-protein coupled receptor 5 (LGR5). The Wnt pathway is involved in cell polarity formation and cell migration, and chlorogenic acid has been found to modulate the Wnt pathway in cancer cells, including colon cancer cells [75]. More specifically, chlorogenic acid has been confirmed to decrease the viability and migratory properties of colorectal cancer cells [76]. As summarized in Table 1, many other reports show that chlorogenic acid exhibits anti-cancer effects by inhibiting cell viability and migratory abilities [76,77], invasion with Akt [78], and ERK [79] inhibition. Chlorogenic acid affects apoptosis by acting on p53, p21, JNK, and nuclear factor erythroid 2-related factor 2 (Nrf2) molecules and also by regulating microRNA expression [80,81,82,83,84]. Furthermore, the combination of multiple phytochemicals is recently emerging as a promising cancer treatment therapy [85]. Chlorogenic acid in combination with cinnamaldehyde and arctigenin exhibited synergistic effects by increasing the number of pathways and systems that can be targeted at once [86].

## 6. Chlorogenic Acid and Neurological Diseases

Chlorogenic acid increases the levels of cyclic adenosine monophosphate (cAMP)-responsive element binding protein (CREB) in the hippocampus and suppresses inflammation in the old brain, facilitating a preventive effect against brain aging [92]. Alzheimer’s disease is the most common progressive neurodegenerative disorder associated with aging. The pathology of this disease is characterized by an earlier accumulation of extracellular amyloid-beta plaques and intracellular neurofibrillary tangles in the hippocampus, eventually leading to severe neurodegeneration, and cognitive and synaptic impairment over time [7]. Additionally, hyperphosphorylation of tau, neuronal inflammation, oxidative stress, and cellular apoptosis can contribute to the pathogenesis of Alzheimer’s disease [7]. Studies have shown that oxidative stress represents a major risk factor associated with the pathology of dementia [70]. More specifically, substantial evidence has confirmed that oxidative stress is associated with neuronal apoptosis and brain dysfunction in Alzheimer’s disease [70]. Due to the absence of actual treatments, brain dysfunction in Alzheimer’s disease is a prevalent public health anxiety. Therefore, a number of preventive factors have been proposed by epidemiological research, including modifiable lifestyle factors, such as healthy dietary habits. In fact, it has been revealed that dietary choices can play a certain role in neuroprotection against the Alzheimer’s disease [93]. However, the relationship between nutrient consumption and neuroprotection is fairly complex. In addition, the convolution of the human diet makes it difficult to examine its distinct effects. Although many lifestyle factors affect brain function, food-related involvements might be a promising strategy in preventing brain dysfunction [93]. Consumption of Arabian coffee containing moderate caffeine seems to ameliorate Alzheimer’s disease-induced cognitive impairment by decreasing amyloid-beta levels [7]. It is also believed that antioxidant nutraceuticals such as coffee phenolic compounds may have beneficial effects in the prevention of Alzheimer’s disease [94].

Parkinson’s disease is a brain disorder that is characterized by neuropsychiatric symptoms such as depression and anxiety preceding the onset of motor symptoms [95]. The major features of this disease include loss of dopaminergic neurons in the substantia nigra and Lewy body depositions [95]. It has been suggested that mitochondrial dysfunction, oxidative stress, and oxidative damage underlie the pathogenesis of Parkinson’s disease [95]. The activity of substantia nigra dopaminergic neurons is critical for striatal synaptic plasticity and associative learning, and degeneration of dopaminergic neurons leads to a disinhibition of the subthalamic nucleus, which in turn increases excitatory projections to the substantia nigra [95]. Environmental exposures to toxic mediators such as ROS may lead to the development of neurodegenerative disorders with similar clinical findings to Parkinson’s disease [95]. Consequently, it is critical to develop strategies to ensure that healthy neurons remain alive following ROS attack without using intricate medications [95]. Epidemiological studies have demonstrated that coffee consumption reduces the risk of Parkinson’s disease, in both case–control and cohort studies, yielding a 33% reduction in risk [96].

## 7. Membrane-Modulating Activity of Chlorogenic Acids

Recently, phenolic compounds have been found to exert modulatory effects in cells through selective action on multiple cell-signaling pathways involved in pathogenesis of degenerative diseases, indicating that the health effects go beyond simple antioxidant activity [43]. The cell membrane of human cells is a mixture of proteins and lipids and forms the boundary between the intracellular compartment and cellular space. With regards to the biological action of chlorogenic acid, current literature demonstrates that chlorogenic acid can alter the biological characteristics of basophil granulocytes by affecting the fluidity of the cell membrane and triggering pseudoallergic reactions [97]. In that study, the authors proposed a mechanism where chlorogenic acid may lead to the aggregation of membrane rafts on the cell membrane surface by altering the fluidity of the cell membrane, thus triggering Syk-related signal transduction and inducing a truncated type I such as an allergic reaction [97]. Another study showed that chlorogenic acid becomes localized mainly in the outer part of the cell membrane, does not induce hemolysis or change the osmotic resistance of erythrocytes, and induces the formation of echinocytes [98]. The values of generalized polarization and fluorescence anisotropy indicate that chlorogenic acid alters the hydrophilic region of the membrane, practically without changing fluidity in the hydrophobic region. The assay of electric parameters showed that chlorogenic acid reduces both the capacity and resistivity of black lipid membranes (BLMs). The overall result is that chlorogenic acid takes position mainly in the hydrophilic region of the membrane, modifying its properties. Such localization allows acids to reduce the concentration of free radicals in the immediate vicinity of the cell and hinder their diffusion into the membrane interior [98]. In addition, caffeic acid displays superficial interactions with cell membrane lipids that can be highly relevant to their biological action [99]. Frias and colleagues reported that chlorogenic acid is a strong phenolic antioxidant with antibacterial properties composed of a caffeoyl ester of quinic acid; however, details on its membrane action and the exact manner with which the composition and membrane state may affect this action, are yet to be fully explored [99]. In their recent study, the interaction of chlorogenic acid with lipid monolayers and bilayers composed by 1,2-di-istoyl-*sn*-glycero-3-phosphocholine (DMPC), 1,2-di-*O*-tetradecyl-*sn*-glycero-3-phosphocholine (14:0 diether PC), 1,2-dipalmitoyl-*sn*-glycero-3-phosphocholine (DPPC), and 1,2-di-*O*-hexadecyl-*sn*-glycero-3-phosphocholine (16:0 diether PC) were investigated at different surface pressures using Fourier transform infrared spectroscopy (FT-IR) measurements [99]. The authors found that the kinetics of interaction was more rapid in DMPC than in the absence of carbonyl groups.

The newly emerging field of membrane lipid therapy involves the pharmacological regulation of membrane lipid composition and structure for the treatment of diseases [100]. Membrane lipid therapy proposes the use of new molecules specifically designed to modify membrane lipid structures and microdomains as pharmaceutical disease-modifying agents by reversing the malfunction or altering the expression of disease-specific protein or lipid signal cascades [100]. As summarized in a semantic review article, the influence of lipids on protein function is reflected in the possibility to use these molecular species as targets for therapies against many diseases and disorders, including inflammation, cancer, and neurodegenerative disorders.

## 8. Conclusions and Future Perspective

As described above, coffee exhibits a variety of positive effects on the immune system by regulating inflammation exhibiting anti-cancer effects and inhibiting the progression of several pathologies of neurodegenerative diseases. In particular, phenolic compounds rich in coffee are considered the main substances that exhibit these effects. Particularly, daily dietary consumption of chlorogenic acids through drinking a cup of coffee has already demonstrated its great potential. Consequently, now that we have harnessed the fragmental evidence of coffee’s beneficial effects in the present review, it is extremely important to establish the deeper knowledge that accounts for the molecular mechanism underlying the beneficial effects of coffee. As a source of anti-inflammatory, anti-cancer and anti-neurodegenerative agents, a cup of coffee holds great promise as a kind of nutraceutical in the pursuit of a healthy human life. Functional foods and their bioactive ingredients are at the interface between nutrition and pharma, and will open the door to seeking new therapeutic intervention for the prevention of diseases [101,102].

## Figures and Tables

**Figure 1 molecules-28-02381-f001:**
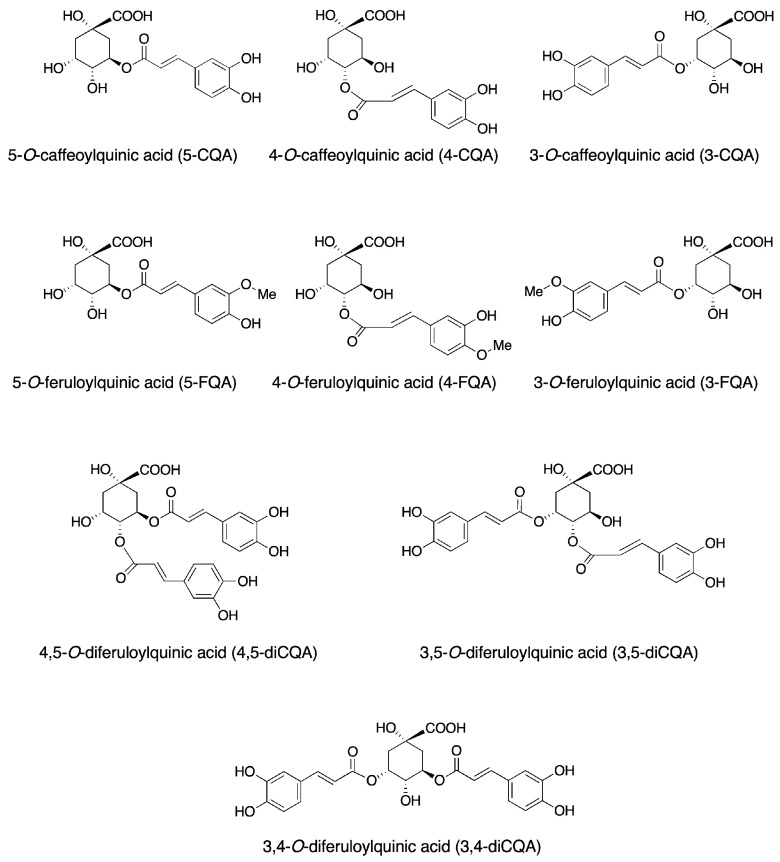
The chemical structure of chlorogenic acids.

**Figure 2 molecules-28-02381-f002:**
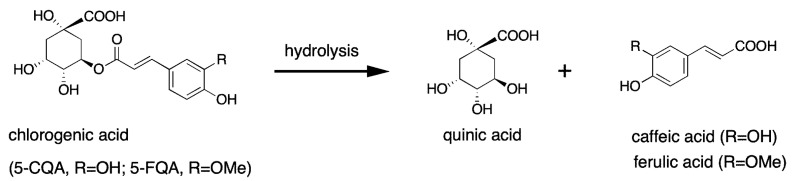
The metabolism of chlorogenic acids.

**Table 1 molecules-28-02381-t001:** The biological effects of CFGAs and the underlying molecular mechanisms.

Effect	Mechanism	Cell	Reference
*Anti-inflammatory effects*			
Adhesion molecule	ICAM-1, VCAM-1	vascular endothelial cells	[48]
Chemotaxis	CD62L	neutrophil	[49]
Leukocyte rolling	CD62P	platelet	[50]
*Anti-cancer effects*			
Proliferation		hepatoma	[56]
Invasion		hepatoma	[57]
MMP activity	MMP-9	hepatoma	[61]
MMP activity	MMP-2	glioma	[63]
Proliferation	PI3K/Akt/mTORC	hepatocellular carcinoma	[73]
Apoptosis	PI3K/Akt/mTORC	kidney cancer	[74]
Signaling	Wnt/β-catenin	colon cancer	[75]
Viability, migration		colorectal cancer	[76]
Migration	DDR1	ovarian cancer	[77]
Invasion	Akt	squamous cell carcinoma	[78]
Invasion	ERK, MMP-2/9	hepatic cancer	[79]
Apoptosis	p53	breast cancer	[80]
Apoptosis	p21	breast cancer	[81]
Apoptosis	JNK	lung cancer	[82]
Apoptosis	Nrf2	hepatocellular carcinoma	[83]
Carcinogenesis	mi-21a-5p	colon cancer	[84]
*Neuroprotective effects*			
Glutamine release	*c*-Src	microglia	[87]
Glutamine release		neuron	[88]
Cell viability		neuron	[89,90]
Neurodegeneration	amyloid-β	neuron	[91]
Brain aging suppression	CREB	microglia	[92]

## Data Availability

There are no data outside that reported in this article.

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
