# Peer review of "The Chemopreventive Effects of Chlorogenic Acids, Phenolic Compounds in Coffee, against Inflammation, Cancer, and Neurological Diseases"

_molecules, 2023, doi:10.3390/molecules28052381_

Round 1

Reviewer 1 Report

1. The phrase "a cup of coffee" appears in both the first and last paragraphs of the text. If this phrase has no special meaning, it is unnecessary to add quotation marks. At the same time, the use of symbols in the text is incorrect. In the same paragraph, the previous phrase uses double quotation marks and then changes to single quotation marks. Please check carefully and modify it.

2. The title of this article is phenolic compounds in coffee. In the full text, the names of other phenolic substances in coffee are mentioned only when describing the chemical composition of coffee. The rest of the full text is about chlorogenic acid, so the title can be more specific.

3. In the article, the first letter of the first word of the sentence is lowercase for many times. For example, in the fourth part of the article, the first letter of chlorogenic acids should be changed to uppercase. Please modify it.

4. At times the English language is awkward and needs to be clarified.

5. The singular and plural nouns in the title of each paragraph are incorrectly used, and the tense of the following paragraphs sometimes contradicts, for example, the last sentence in Part IV, please carefully check and modify.

6. Whether the other phenolic compounds in coffee, except chlorogenic acid, have several activities related to the topic. More specifically, whether the other phenolic compounds mentioned in the article also participate in anti-inflammatory and anti-cancer activities, if any, should be supplemented in the corresponding part.

7. The narrative mode of this article is too simple. When describing the anticancer activity of chlorogenic acid, some drug target maps and action mechanisms can be appropriately added for easy reading.

Author Response

Reviewer1

  1. The phrase "a cup of coffee" appears in both the first and last paragraphs of the text. If this phrase has no special meaning, it is unnecessary to add quotation marks. At the same time, the use of symbols in the text is incorrect. In the same paragraph, the previous phrase uses double quotation marks and then changes to single quotation marks. Please check carefully and modify it.

We thank the reviewer for carefully reading the manuscript. The quotation marks have been removed (page 1, line 30; page 9, line 5; page 9, line 8).

  1. The title of this article is phenolic compounds in coffee. In the full text, the names of other phenolic substances in coffee are mentioned only when describing the chemical composition of coffee. The rest of the full text is about chlorogenic acid, so the title can be more specific.

Following the suggestion, we modified the title. 

  1. In the article, the first letter of the first word of the sentence is lowercase for many times. For example, in the fourth part of the article, the first letter of chlorogenic acids should be changed to uppercase. Please modify it.

We thank the reviewer for carefully reading our manuscript. The error has been corrected.

  1. At times the English language is awkward and needs to be clarified.

The manuscript has been modified accordingly.

  1. The singular and plural nouns in the title of each paragraph are incorrectly used, and the tense of the following paragraphs sometimes contradicts, for example, the last sentence in Part IV, please carefully check and modify.

We thank the reviewer for carefully reading our manuscript. The error has been corrected.

  1. Whether the other phenolic compounds in coffee, except chlorogenic acid, have several activities related to the topic. More specifically, whether the other phenolic compounds mentioned in the article also participate in anti-inflammatory and anti-cancer activities, if any, should be supplemented in the corresponding part.

Coffee beans as well as its brew are the richest source of phenolic compounds. Among them predominant are chlorogenic acids, although other phenolic compounds exist in coffee. Therefore, in this article, we focused on coffee chlorogenic acids as the main constituent that exhibit a variety of beneficial effects on health. Accordingly, we modified the title of this manuscript by adding “chlorogenic acids.” I sincerely appreciate your consideration.

  1. The narrative mode of this article is too simple. When describing the anticancer activity of chlorogenic acid, some drug target maps and action mechanisms can be appropriately added for easy reading.

The manuscript has been modified accordingly.

Reviewer 2 Report

The manuscript (ID: molecules-221767) reviews the positive effect of phenolic compounds, especially chlorogenic acids, from coffee on several diseases, such as cancer, inflammation, and neurodegenerative diseases. Due to its beneficial biological effects on the human body, coffee is considered a functional food. The authors summarized the literature data about phytochemicals contained in coffee as nutraceuticals with a focus on the reduction of disease risk and its prevention. The study is well-designed to meet the goals. It is obviously challenged by the types of data accumulated.

The authors are recommended to address the following comments during the revision:

1.     The manuscript should be Grammarly corrected. The recommended corrections have been noticed in the pdf file of the manuscript.

2.     Is the mechanism behind the role of chlorogenic acid in the prevention of neurodegenerative diseases known? If yes, complete Section 6.

3.     The study mainly considers the positive role of coffee compounds. However, it is known that the consumption of coffee also has negative effects, so they should be mentioned.

4.     I suggest reducing the number of references.

5.     According to the reference list, the authors have not participated in any research related to coffee and its compounds, so why they decided to write a review on that topic? Undisputed competence is in the field dealing with the research of numerous diseases described in the manuscript.

Author Response

Reviewer2

The manuscript (ID: molecules-221767) reviews the positive effect of phenolic compounds, especially chlorogenic acids, from coffee on several diseases, such as cancer, inflammation, and neurodegenerative diseases. Due to its beneficial biological effects on the human body, coffee is considered a functional food. The authors summarized the literature data about phytochemicals contained in coffee as nutraceuticals with a focus on the reduction of disease risk and its prevention. The study is well-designed to meet the goals. It is obviously challenged by the types of data accumulated.

The authors are recommended to address the following comments during the revision:

We thank the reviewer for carefully reading our manuscript and providing us constructive comments to improve the manuscript.

  1. The manuscript should be Grammarly corrected. The recommended corrections have been noticed in the pdf file of the manuscript.

The manuscript has been modified accordingly.

  1. Is the mechanism behind the role of chlorogenic acid in the prevention of neurodegenerative diseases known? If yes, complete Section 6.

Epidemiological studies have associated coffee consumption with beneficial effects including reduced instances of neurodegenerative diseases including Alzheimer's disease and Parkinson's disease. These beneficial effects might be attributed to antioxidative feature of chlorogenic acid, but the mechanism underlying the pharmacokinetics of coffee chlorogenic acids remain unelucidated. Further studies will be needed to elucidate the action mechanism responsible for those effect, and thus we do not mention the mechanism behind the role of chlorogenic acid in the prevention of neurodegenerative diseases at the moment. I appreciate your consideration.

  1. The study mainly considers the positive role of coffee compounds. However, it is known that the consumption of coffee also has negative effects, so they should be mentioned.

The negative effects of coffee consumption are mentioned on page 2 of the revised manuscript (lines21-24); “However, caffeine intake also has negative effects although the amount of caffeine in a cup of coffee is influenced by the method of coffee preparation such as boiled coffee, filtered coffee, and espresso, and the half-life of caffeine in human body is approximately 4–6 h (26).”  We thank the reviewer for critical reading of the manuscript.

  1. I suggest reducing the number of references.

Following the reviewer’s advice, the references that relate to the membrane structure, no. 97-102 and 105, has been deleted from the previous version of the manuscript. The references no. 108 and 109 have also been deleted.

  1. According to the reference list, the authors have not participated in any research related to coffee and its compounds, so why they decided to write a review on that topic? Undisputed competence is in the field dealing with the research of numerous diseases described in the manuscript.

The reason why we decided to write a review on this topic is that we have been engaged in the research on contribution of coffee to improve of the health of the people. This work was supported in part by The All Japan Coffee Association, an organization to unify the coffee industry in Japan established in 1980, but we have no conflict of interest relevant to this article. These have been described at the end of the revised manuscript (page 12, lines 2 and 6).

Reviewer 3 Report

1.     Language of the manuscript needs much improvement and there are many grammatical mistakes in the manuscript.

2.     Please provide more information on action mechanisms of the phenolic compounds.

3.     Abbreviations should be avoided when they are mentioned for the first time. Please check the full text.

4.     Why certain phenolic compounds are selected as examples to be described in the text? Are they more effective or with other reasons?

5.     The introduction of the background of phenolic compounds from coffee in inflammation, cancer, and neurological diseases is insufficient.

6.     This paper only reviews the existing studies, but does not put forward the existing problems and research prospects.

Author Response

Reviewer3

  1. Language of the manuscript needs much improvement and there are many grammatical mistakes in the manuscript.

The manuscript has been modified accordingly.

  1. Please provide more information on action mechanisms of the phenolic compounds.

We appreciate the advice. Accordingly, we described more information on action mechanisms of the phenolic compounds (page 3, line30-page 4, line 4). We also added a sentence on action mechanism of the phenolic compounds that might go beyond simple antioxidant activity (page 7, lines 34-36). This prompted us to elaborate on the membranal action of the phenolic compounds in section 7.

  1. Abbreviations should be avoided when they are mentioned for the first time. Please check the full text.

The error has been corrected. The abbreviations used are listed in the revised manuscript (pages 10-11).

  1. Why certain phenolic compounds are selected as examples to be described in the text? Are they more effective or with other reasons?

The reason why we decided to write a review on this topic is that we have been engaged in the research on contribution of coffee to improve of the health of the people. This work was supported in part by The All Japan Coffee Association, but we have no conflict of interest relevant to this article. These have been described at the end of the revised manuscript (page 12, lines 2 and 6).

  1. The introduction of the background of phenolic compounds from coffee in inflammation, cancer, and neurological diseases is insufficient.

We agree to the comments. Accordingly, we added the description concerning the relation between coffee consumption and neurological diseases and inflammation (page 1, line 35-page 2, line 5).

  1. This paper only reviews the existing studies, but does not put forward the existing problems and research prospects.

We thank the Reviewer for this comment. Following the advice, we added a sentence to the Conclusion and Future Perspective section to put forward the research prospects by citing two very important papers (reference no.101 and 102 of the revised manuscript).

Round 2

Reviewer 3 Report

None

Author Response

Thank you so much for the good evaluation on our manuscript.